# Enterocytes, fibroblasts and myeloid cells synergize in anti-bacterial and anti-viral pathways with IL22 as the central cytokine

Jean Paul ten Klooster [1✉], Marianne Bol-Schoenmakers[2], Kitty van Summeren[1], Arno L. W. van Vliet[3], Cornelis A. M. de Haan[3], Frank J. M. van Kuppeveld [3], Saertje Verkoeijen[1] & Raymond Pieters[1,2]

IL22 is an important cytokine involved in the intestinal defense mechanisms against microbiome. By using ileum-derived organoids, we show that the expression of anti-microbial peptides (AMPs) and anti-viral peptides (AVPs) can be induced by IL22. In addition, we identified a bacterial and a viral route, both leading to IL22 production by T cells, but via different pathways. Bacterial products, such as LPS, induce enterocyte-secreted SAA1, which triggers the secretion of IL6 in fibroblasts, and subsequently IL22 in T cells. This IL22 induction can then be enhanced by macrophage-derived TNFα in two ways: by enhancing the responsiveness of T cells to IL6 and by increasing the expression of IL6 by fibroblasts. Viral infections of intestinal cells induce IFNβ1 and subsequently IL7. IFNβ1 can induce the expression of IL6 in fibroblasts and the combined activity of IL6 and IL7 can then induce IL22 expression in T cells. We also show that IL22 reduces the expression of viral entry receptors (e.g. ACE2, TMPRSS2, DPP4, CD46 and TNFRSF14), increases the expression of anti-viral proteins (e.g. RSAD2, AOS, ISG20 and Mx1) and, consequently, reduces the viral infection of neighboring cells. Overall, our data indicates that IL22 contributes to the innate responses against both bacteria and viruses.

[1] Research Centre Healthy and Sustainable Living, Innovative Testing in Life Sciences and Chemistry, University of Applied Sciences Utrecht, Utrecht, The Netherlands. [2] Institute for Risk Assessment Sciences, Population Health Sciences Division, Utrecht University, Utrecht, The Netherlands. [3] Virology Section, Infectious Disease and Immunology Division, Department of Biomolecular Health Sciences, Faculty of Veterinary Medicine, Utrecht University, Utrecht, The Netherlands. ✉email: jeanpaul.tenklooster@hu.nl

The intestine contains many bacteria which, when given the opportunity, cause local damage and invade the mucus layer and the underlying tissues[1]. Antibacterial peptides, such as Reg3β and Reg3γ, produced by Paneth cells in the small intestine, ensure that the inner intestinal mucus layer contains little to no bacteria[2]. The importance of the C-type lectin Reg3γ in this process has been demonstrated by Hooper et al., who show that Reg3γ knockout mice suffer from a high influx of bacteria in the intestinal mucus layer[2]. In addition, lack of Reg3γ expression in human subjects is highly associated with Inflammatory Bowel Disease (IBD)[1,3,4].

The expression of Reg3γ can be enhanced by IL22, a cytokine that can be secreted by neutrophils, Th22, Th17, ILC3, or γδ T cells, but the regulation of IL22 expression and secretion in these different cell types is not entirely clear[5–8]. There is data suggesting that the cytokines IL23, IL6, TNFα, and IL1α/β and Serum Amyloid A3 (SAA3) are involved in the induction of IL22 in T cells of different origin[5–8]. These cytokines can be induced in myeloid cells, such as macrophages and dendritic cells (DC) by LPS[9], a potent activator of the TLR4 pathway[10]. The combination of IL6, IL1β, and TNFα can induce the expression of IL22 in adult peripheral blood cells when combined with anti-CD3/anti-CD28[5,6]. SAA3 can induce IL22 expression in bone marrow-derived neutrophils[7] and in in vitro-differentiated Th17 cells[11].

In addition to bacteria, the gut is also home to a diverse population of viruses. Most of these viruses are bacteriophages, which target bacteria and thus affect the bacterial composition in the intestine[12]. Besides bacteriophages, the gut also hosts viruses that target eukaryotes, such as Hepeviridae, Polydnaviridae, Tymoviridae, and Virgaviridae families, which have also been associated with the pathogenesis of IBD[13–15].

Our data demonstrates a previously unrecognized pathway of bacterial and viral induced IL22 production by intestinal T cells. By using co-cultures of 2D and 3D intestinal organoids, fibroblasts, macrophages, DC and T lymphocytes, we show that IL6 and TNFα synergize in stimulating the production of IL22 in T cells when stimulated with bacterial products, such as LPS, Pam2, and Pam3. In addition, we show that viral infections of intestinal organoids induce IFNβ1 that, via subsequent autocrine stimulation, induces IL7 expression in enterocytes. Moreover, the secreted IFNβ1 can induce the expression of IL6 in fibroblasts and the combined secretion of IL6 and IL7 can also induce the expression of IL22 in T cells, and thus reduce the expression of viral entry receptors (e.g., ACE2, TMPRSS2, DPP4, TNFRSF14, and CD46) and enhance the expression of anti-viral genes (e.g., RSAD2, ISG20, and AOS).

## Results

### A DC/macrophage-T cell axis is required for LPS-induced expression of Reg3β and Reg3γ in ileum-derived organoids.

Recently, it has been suggested that Paneth cells can be directly stimulated by LPS to produce Reg3γ[2]. However, these experiments were performed in vivo and could not exclude the role of immune cells, which are very likely to be involved in this process[5,7,11]. To test whether LPS directly induces expression of Reg3γ in Paneth cells or requires the mediation of immune cells, we stimulated the dendritic JAWSII cells and macrophage RAW264.7 cells with LPS and used the supernatants of these cells to either stimulate EL4 T cells or ileum-derived organoids. Organoids were then analyzed for Reg3β and Reg3γ mRNA expression (Fig. 1a). We observed that the expression of these genes in organoids was not stimulated directly by LPS, but rather by supernatants of EL4 cells pre-exposed to the supernatant of LPS-activated JAWSII or RAW264.7 cells (Fig. 1a). This suggests that the activated DCs or macrophages produce a soluble factor that activates EL4 T cells to produce another factor, that, subsequently, can activate Paneth cells to produce Reg3β or Reg3γ.

### Multiple bacterial products induce Reg3β and Reg3γ expression in Paneth cells.

LPS is the most studied bacterial product with respect to its effect on Reg3β and Reg3γ expression by Paneth cells. However, there are many other bacterial products which can activate dendritic cells or macrophages[16] and possibly stimulate Paneth cells to produce Reg3β and Reg3γ through DC and T cell activation. To test this, we stimulated EL4 T cells with supernatant derived from Pam2 and Pam3-stimulated JAWSII cells and subsequently stimulated organoids with these EL4 supernatants. Indeed, we observed that mRNA expression of Reg3β and Reg3γ was also induced by Pam2 and Pam3, although not as efficiently as by LPS (Fig. 1b). This suggests that the induction of the Reg3 genes is mediated via a general DC activation pathway and is not restricted to LPS.

### IL22 and IFNγ induce the expression of Reg3β and Reg3γ in Paneth cells.

The fact that the supernatant of activated EL4 cells

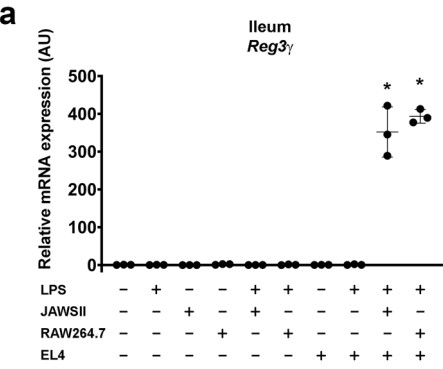

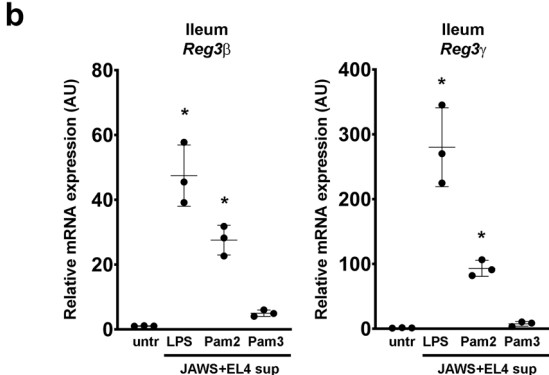

**Fig. 1 Reg3β and Reg3γ expression in ileum-derived organoids is dependent on LPS-activated RAW264.7 macrophages or JAWSII dendritic cells and EL4 T cells. a** RAW264.7 cells and JAWSII cells stimulated with LPS (1 µg/ml) for 24 h and subsequently, supernatants were first incubated with EL4 T cells for 24 h and then added to organoids or they were added to organoids directly. Reg3γ mRNA was determined by QPCR and normalized to untreated organoids. **b** JAWSII cells stimulated with either LPS, Pam2, or Pam3 (1 µg/ml) for 24 h and subsequently, supernatants were first incubated with EL4 T cells for 24 h and then added to organoids. Reg3β and Reg3γ mRNA was determined by QPCR and normalized to untreated organoids. All experiments were performed in triplicate with a minimum of three independent experiments. Data are shown as mean ± SD. For statistical analyses, Log2 transformed data were used in Welch and Brown–Forsythe tests followed by Dunnett's T3 multiple comparisons test. *P < 0.05 was considered to indicate statistical significance.

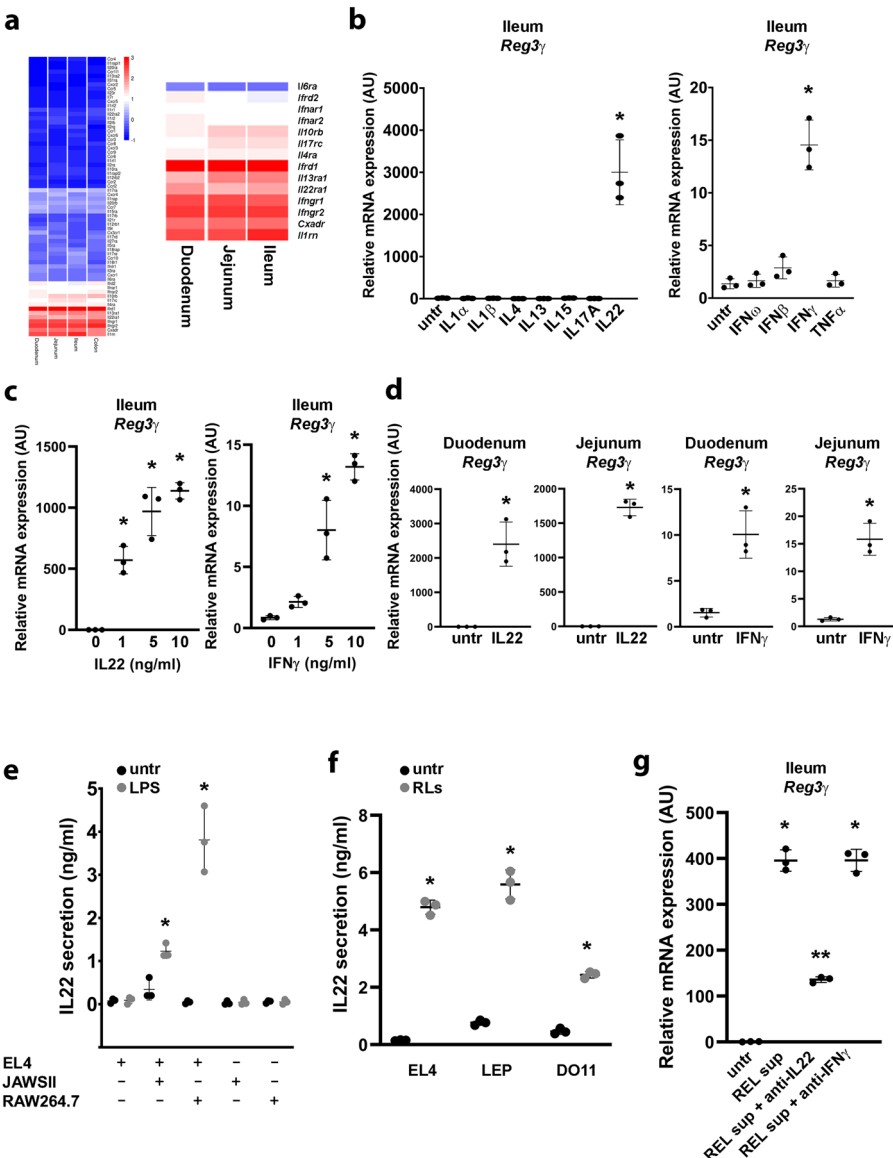

**Fig. 2 Reg3γ expression in intestinal organoids is induced by T cell-derived IL22 or IFNγ. a** Cytokine receptor mRNA expression in duodenum, jejunum and ileum-derived organoids. **b** *Reg3γ* mRNA expression in ileum-derived organoids after 24 h exposure to cytokines (10 ng/ml). **c** IL22-induced and IFNγ-induced mRNA expression of *Reg3γ* in ileum-derived organoids. **d** IL22- and IFNγ-induced mRNA expression of *Reg3γ* in duodenum-derived or jejunum-derived organoids. **e** ELISA for IL22 on supernatants from EL4 T cells with prior exposure to supernatants from LPS-stimulated RAW264.7 or JAWSII cells. **f** ELISA for IL22 on supernatants from EL4, LEP, or DO11 T cells which were exposed to supernatants from LPS-stimulated RAW264.7 cells (RLs) for 48 h. **g** *Reg3γ* mRNA expression in ileum-derived organoids after exposure of EL4-derived supernatants with prior exposure to LPS-treated RAW264.7 cell supernatant (REL) in the presence or absence of blocking antibodies for IL22 or IFNγ. All experiments were performed in triplicate with a minimum of three independent experiments. Data are shown as mean ± SD. For statistical analyses (**a–c**, **e**, and **g**), Log2 transformed data were used in Welch and Brown–Forsythe tests followed by Dunnett's T3 multiple comparisons test. For comparison in **d** and **f**, we performed an unpaired *t*-test (two-tailed) with Welch's correction on Log2 transformed data.*P < 0.05 was considered to indicate statistical significance.

can stimulate *Reg3β* and *Reg3γ* expression in organoids, indicates that a soluble factor is involved, most likely a cytokine. To test which cytokine(s) can activate Paneth cells, we performed a microarray on ileum-derived organoids and noticed that these organoids express only a small subset of cytokine receptors (Fig. 2a and Fig. S1). To test which of these receptors can eventually lead to *Reg3β* and *Reg3γ* expression, we stimulated ileum-derived organoids with the recombinant cytokines IL4, IL13, IL15, IL17, IL22, IFNω, IFNβ, IFNγ, and TNFα. These experiments revealed that of these cytokines only IL22 and IFNγ induce the expression of *Reg3β* and *Reg3γ* (Fig. 2b) in a concentration-

dependent manner (Fig. 2c) and that this is also true for duodenum- and jejunum-derived organoids (Fig. 2d).

Next, we performed an ELISA for IL22 and IFNγ to determine which of these two cytokines were produced in EL4 cells after exposure to LPS-treated JAWS II or RAW264.7 supernatants and observed that while IL22 is produced (Fig. 2e), IFNγ is not produced by EL4 cells (Fig. S2). Similarly, two other T cell lines, DO11 and LEP, also produced IL22 (Fig. 2f) but no IFNγ (Fig. S2) after exposure to LPS-treated RAW264.7 supernatants. Of note, these T cells can all produce IFNγ when the TCR is activated by using anti-CD3 (Fig. S2). To confirm that IL22 is indeed the

 COMMUNICATIONS BIOLOGY | https://doi.org/10.1038/s42003-021-02176-0

soluble factor that mediates the induction of *Reg3γ* in organoids, we treated organoids with EL4-derived supernatant, with prior RAW264.7-LPS supernatant exposure, in the absence or presence of blocking antibodies for IL22 and IFNγ (Fig. 2g) and their receptors (Fig. S3). Indeed, blocking IL22 signaling resulted in a lower expression of *Reg3γ* (Fig. 2g), confirming that IL22 mediates the induction of *Reg3γ* in intestinal organoids by LPS-activated myeloid cells and T cells.

**Intestinal epithelial lymphocytes produce both IL22 and IFNγ after exposure to LPS-stimulated dendritic cell-derived supernatants.** To confirm that our in vitro systems reflects the in vivo situation, we isolated intestinal epithelial lymphocytes (IELs) from a C57BL/6 mouse and exposed these cells to supernatant from LPS-activated bone marrow-derived DCs. IELs were cultured in the absence of other activating factors, such as anti-CD3 or IL7. We determined *Il22* and *Ifnγ* expression by QPCR (Fig. 3a) and ELISA (Fig. 3b). Indeed, these primary cells also produced IL22 after exposure to the DC-LPS supernatant, similar to the EL4 cells. However, we also observed a significant induction of IFNγ in these cells (Fig. 3a, b). Subsequently, organoids were exposed to IEL supernatants for 24 h in the absence or presence of blocking antibodies for IL22 and IFNγ and we analyzed for *Reg3β*, *Reg3γ*, and *Cd74* mRNA expression by QPCR (Fig. 3c). The expression of the IFNγ-target gene *Cd74* was included to test for IFNγ activity (Fig. 3c). Similar to the EL4 experiments in Figs. 1 and 2, we observed that blocking IL22 was sufficient to inhibit the induction of the *Reg3* genes (Fig. 3c), whereas blocking IFNγ enhanced the expression of *Reg3β* and *Reg3γ*, suggesting that IFNγ rather inhibits IL22-induced expression of these *Reg3* genes. This was confirmed by comparing the expression of *Reg3γ* in organoids treated with recombinant IL22 or a mixture of IL22 and IFNγ for 24 h (Fig. 3d), showing that the presence of IFNγ clearly results in lower *Reg3γ* induction by IL22.

**IL6/TNFα or IL6/IL7 induce expression of IL22 in EL4 T cells.** Research by others demonstrates that the cytokines IL23, IL6, TNFα, and IL1α/β may be involved in the induction of IL22 in T cells[5,7,11]. To test the involvement of these cytokines in our in vitro system, we analyzed our previously published microarray dataset on RAW264.7 cells which were stimulated with LPS[17] and observed the expression of multiple cytokines, including *Il23*, *Il6*, *Tnfα*, and *Il1α/β* (Fig. 4a). Next, we exposed EL4 cells to these cytokines and included recombinant cytokines that were also upregulated in RAW264.7 cells after LPS treatment, and tested the secretion of IL22 using HEK-Blue IL22-reporter cells (Fig. 4b). In these experiments, IL9 was included as a positive control for IL22 induction[18]. We observed that IL6 induced IL22 secretion in EL4 cells at a concentration of 50 ng/ml (Fig. 4b) and obviously less efficient at 10 ng/ml (Fig. 4c). When this less efficient concentration of IL6 was combined with other cytokines, we observed that IL1β, IL7, or TNFα synergized with IL6 to induce IL22 secretion in EL4 cells (Fig. 4c, d). Using capturing antibodies for IL1β, IL6, IL7, IL9, IL23, and TNFα, only those against IL6, TNFα and, to lesser extent, IL1β, were able to reduce IL22 secretion by EL4 T cells stimulated with supernatants from RAW264.7-LPS (Fig. 4e). Enhanced IL22 secretion induced by the combination of IL6 and TNFα was also observed in freshly isolated IELs that were stimulated for 48 h with IL6 and TNFα (Fig. 4f). This suggests that IL6 and TNFα synergize to induce IL22 secretion.

**Fibroblasts enhance the LPS-macrophage-induced induction of IL22 in EL4 cells by increasing IL6 expression.** In the intestine,

macrophages and DCs present a well-known source of IL6[19]. However, fibroblasts are also known for their ability to produce IL6[20]. Therefore, we determined the role of fibroblasts in the production of IL22 by T cells. To this end, we tested different co-culture combinations of NIH3T3 fibroblasts, RAW264.7 macrophages and EL4 T cells and analyzed the IL22 secretion after 24 h of LPS treatment (Fig. 5a). We observed a strong IL22 production when NIH3T3, RAW264.7 and EL4 were combined in cultures that were incubated with LPS (Fig. 5a). This response was much higher than observed in LPS-treated co-cultures of RAW264.7 and EL4 without fibroblasts (Fig. 5a), even when cell concentration of RAW264.7 and EL4 co-cultures were increased ten times (Fig. 5a).

Appreciating the importance of IL6 and TNFα in the induction of IL22, we tested whether the increased IL22 response of these co-cultures was due to an increased IL6 or TNFα secretion. By performing an ELISA on supernatants of co-cultures of RAW264.7 and NIH3T3 cells (Fig. 5b, c), we found that the production of TNFα was not altered when comparing co-cultures to single cultures (Fig. 5c). However, IL6 secretion was greatly enhanced in the co-cultures after LPS stimulation (Fig. 5b). The induction of IL6 in RAW264.7 cells is dependent on NFκB[16]. By using a RAW264.7 NFκB-reporter in the co-cultures with fibroblast, we demonstrated that the NFκB-activity in RAW264.7 cells did not change after LPS stimulation when compared to RAW264.7 single cultures (Fig. 5d), suggesting that the increased IL6 was produced by the fibroblasts.

To further test if TNFα can induce IL6 in fibroblasts, we stimulated NIH3T3 cells with RAW264.7-LPS supernatant (RLs) with or without a capturing antibody for TNFα and determined the IL6 secretion by ELISA (Fig. 5e). We observed that the induction of IL6 by RAW264.7-LPS supernatant was inhibited by blocking TNFα. Moreover, the addition of TNFα to NIH3T3 fibroblasts induced mRNA expression of *Il6*, but also of *Il1α* (Fig. 5f). In addition, IL1α treatment also induced IL6 expression in NIH3T3 cells (Fig. 5g). The TNFα-induced expression of IL6 after 2 days was inhibited by a blocking antibody for IL1α, suggesting that the induction of IL6 by TNFα in fibroblasts after 2 days is partially regulated by the induction of IL1α by TNFα (Fig. 5g). Overall, this means that both IL1α and TNFα co-induce IL6 secretion in fibroblasts, and that TNFα-mediated IL1α autocrine stimulation enhances the TNFα-induced expression of IL6 in fibroblasts.

**LPS-induced expression of SAA3 in organoids stimulates IL22 secretion in EL4 cells through IL1α expression in RAW264.7 macrophages and IL6 expression in NIH3T3 fibroblasts.** Bacterial components such as LPS, Pam2 and Pam3 can activate TLR4 signaling in various cell types, and can induce a strong pro-inflammatory response in macrophages and dendritic cells[9]. However, intestinal enterocytes are constantly exposed to these bacterial compounds, whereas the intestine is not constantly highly inflamed, suggesting that intestinal epithelial cells (IECs) do not produce high levels of pro-inflammatory cytokines or might express anti-inflammatory cytokines, which could inhibit the inflammatory activity of intestinal myeloid cells. To test which genes are activated in intestinal enterocytes in response to LPS, we seeded ileum-derived organoids either in Matrigel (3D) or on collagen I-coated 48 wells plates (2D)[21] and allowed them to grow for 2 days. The different culture systems, 2D vs. 3D, allow us to discriminate between crypt structures (3D), enriched in stem cells and Paneth cells, and more differentiated villi-like cultures (2D)[9], containing high amounts of enterocytes and endocrine cells. Subsequently, we stimulated these cultures with LPS, Pam2, or Pam3 for 24 h. From these cultures we isolated

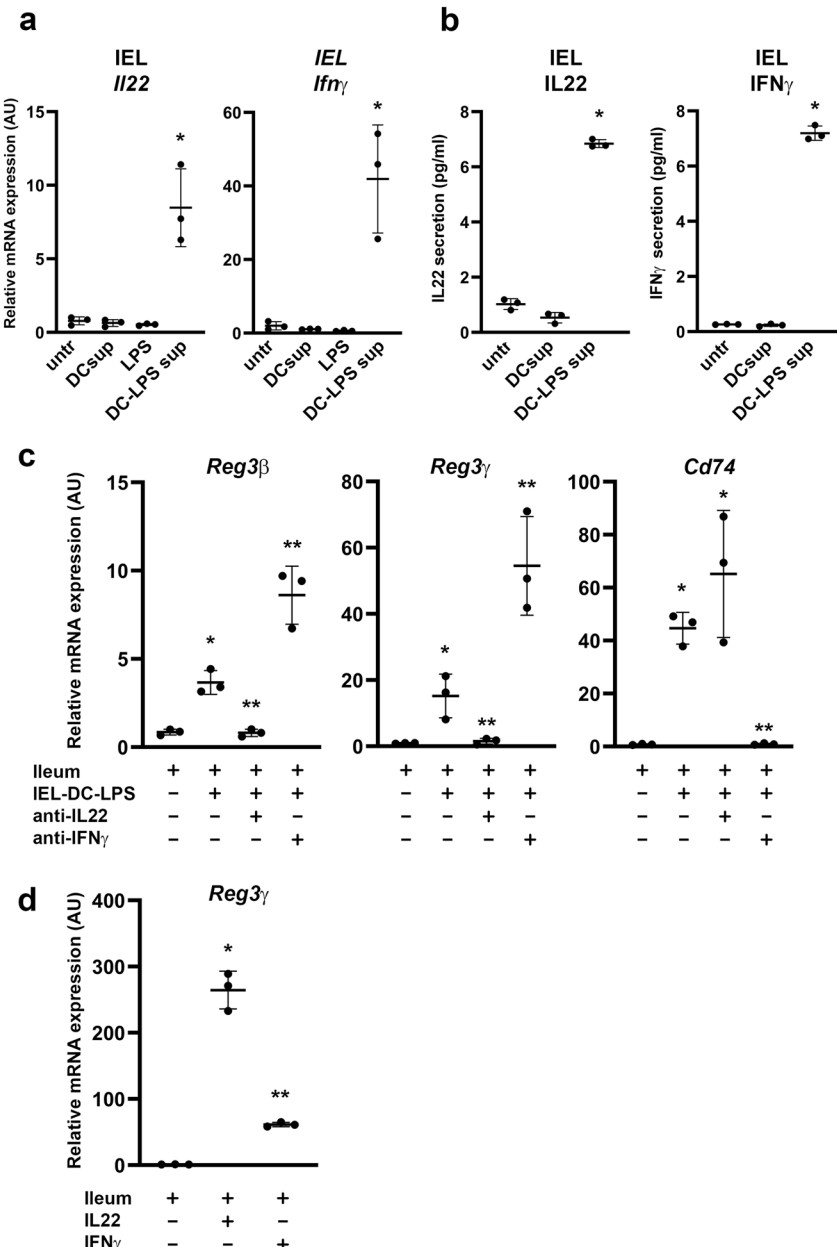

**Fig. 3 Reg3γ expression in intestinal organoids is induced by IEL-derived IL22 or IFNγ. a** *Il22* and *Ifnγ* mRNA expression in intestinal epithelial lymphocytes, freshly isolated from mouse small intestine, and stimulated with supernatants form LPS-treated bone marrow-derived dendritic cells. **b** Protein secretion by intestinal epithelial lymphocytes, freshly isolated from mouse small intestine, and stimulated with supernatants form LPS-treated bone marrow-derived dendritic cells. **c** *Reg3β, Reg3γ*, and *Cd74* mRNA expression in ileum-derived organoids after 24 h exposure to supernatants from IEL cells with prior exposure to supernatants from LPS-stimulated BMDCs in the presence or absence of blocking antibodies for IL22 or IFNγ. **d** *Reg3γ* mRNA expression in ileum-derived organoids after 24 h exposure to IL22 alone or IL22 combined with IFNγ (5 ng/ml). All experiments were performed in triplicate with a minimum of three independent experiments. All data are shown as mean ± SD. For statistical analyses, Log2 transformed data were used in Welch and Brown–Forsythe tests followed by Dunnett's T3 multiple comparisons test. *P < 0.05 was considered to indicate statistical significance. **P < 0.05 compared to stimulated samples.

RNA to perform microarray analyses. Data analysis on genes involved in inflammatory pathways, revealed that *Tnfsf13/April*, *Saa3*, and *Tnfα* were upregulated after LPS stimulation of organoids cultured in 2D (Fig. 6a). We also found that stimulation of 2D cultures with the bacterial compound Pam3 increased the expression of *Saa3* to similar levels as LPS (Fig. 6b). The 2D cultures are highly enriched in enterocytes[9] and indeed, using the human enterocyte cell line HT29, we confirmed that LPS induces the expression of *Saa1*, the human homolog of mouse *Saa3* (Fig. 6c).

SAA3 can activate NFκB signaling[22] and therefore, we set out to identify the effect of SAA3 on *Il6*, *Il1α*, and *Tnfα* production in RAW264.7 macrophages (Fig. 6e) and *Il6* and *Il1α* production in NIH3T3 fibroblasts (Fig. 6f, g) and observed an increase in *Il1α* in RAW264.7 cells and an increase of *Il1α* and *Il6* in NIH3T3 cells (Fig. 6e–g). This suggests that SAA3 can induce IL6 expression in fibroblasts directly or indirectly through macrophage IL1α secretion.

Based on our observation that IL6 can induce the expression of IL22 in EL4 T cells, we would expect that SAA3/1 can induce

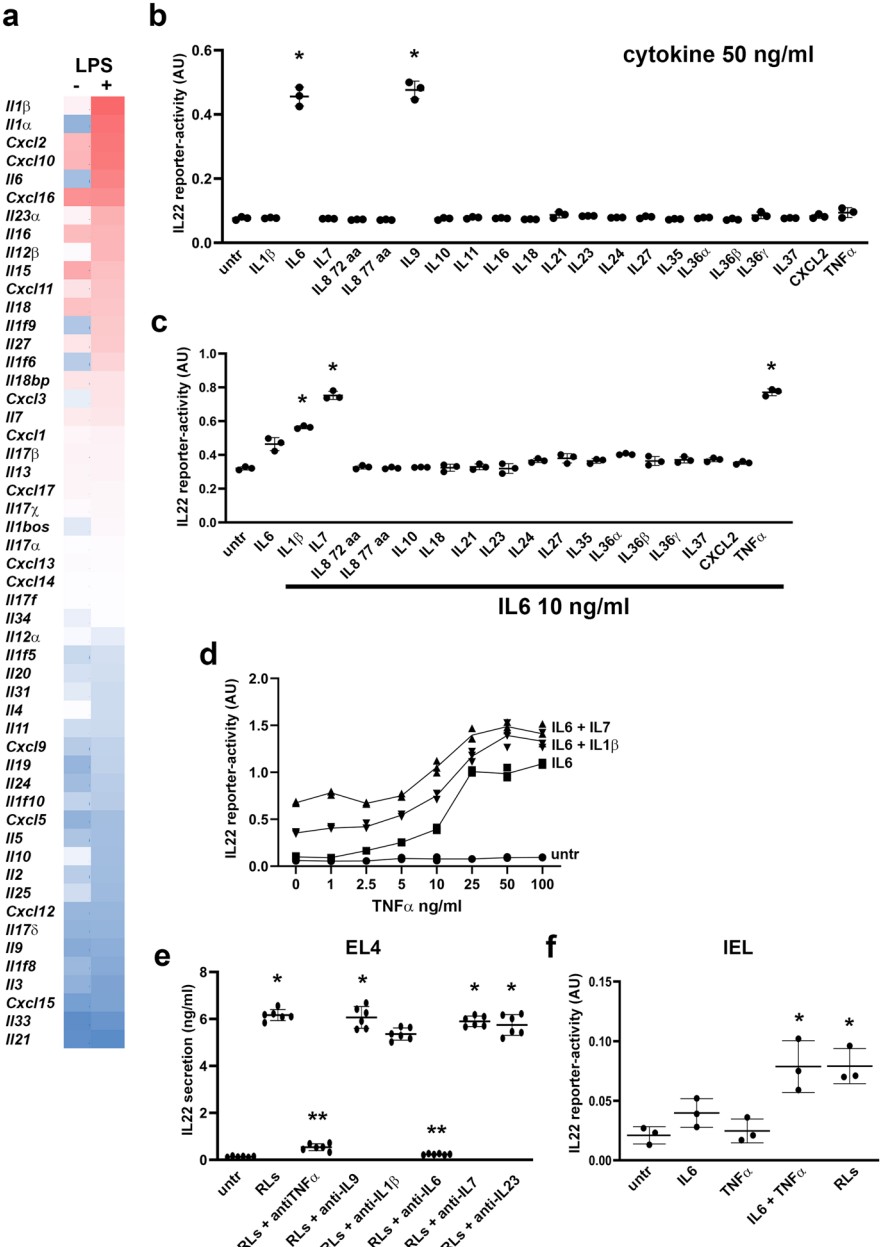

**Fig. 4 IL22 secretion and expression in EL4 T cells is induced by IL6 and enhanced by either IL7 or TNFα and/or IL1β. a** Cytokine mRNA expression in LPS-stimulated RAW264.7 cells. **b** IL22 reporter-activity of supernatants from EL4 T cells stimulated with indicated cytokines at concentrations of 50 ng/ml. **c** IL22 reporter-activity of supernatants from EL4 T cells stimulated with IL6 combined with indicated cytokines at concentrations of 10 ng/ml. **d** IL22 reporter-activity of supernatants from EL4 T cells stimulated with IL6 (10 ng/ml) combined with either IL1β (10 ng/ml) or IL7 (5 ng/ml) and TNFα (with indicated concentrations). **e** IL22 expression in EL4 T cells after 48 h exposure to supernatants from LPS-stimulated RAW264.7 cells (RLs) in the presence or absence of indicated blocking antibodies. **f** IL22 reporter-activity of supernatants from freshly isolated IEL cells stimulated with either IL6 or TNFα, or IL6 and TNFα combined (10 ng/ml), or stimulated with supernatant from LPS-stimulated RAW264.7 cells. All experiments were performed in triplicate with a minimum of three independent experiments. All data are shown as mean ± SD. For statistical analyses, Log2 transformed data were used in Welch and Brown–Forsythe tests followed by Dunnett's T3 multiple comparisons test. *$P < 0.05$ was considered to indicate statistical significance. **$P < 0.05$. compared to RLs stimulated samples.

IL22 in EL4 T cells mediated by myeloid cells and fibroblasts. We tested this, by exposing EL4 T cells to the supernatants of untreated and hSAA1-treated RAW264.7 cells, NIH3T3 cells and RAW264.7/NIH3T3 co-cultures and determined the secretion of IL22 (Fig. 6h). In agreement with IL6 being present in 3T3 cultures after hSAA1 or IL1α exposure, we only found IL22 secretion in EL4 cells that were exposed to supernatants from fibroblasts or fibroblast/macrophage co-cultures stimulated with hSAA1 (Fig. 6h), which could be blocked by an inhibitory

anti-IL6 antibody (Fig. 6i). This indicates that EL4 T cells cannot be directly activated by SAA3 and require IL6 from fibroblasts to induce IL22 expression.

**Viral-induced IL7 from IECs enhances IL6-mediated production of IL22.** In Fig. 4 we show that IL7 can enhance IL6-mediated IL22 secretion by EL4 T cells and next, we tested whether there is a direct concentration–response relationship between IL-7 and IL-6-induced IL-22 production (Fig. 7a). While

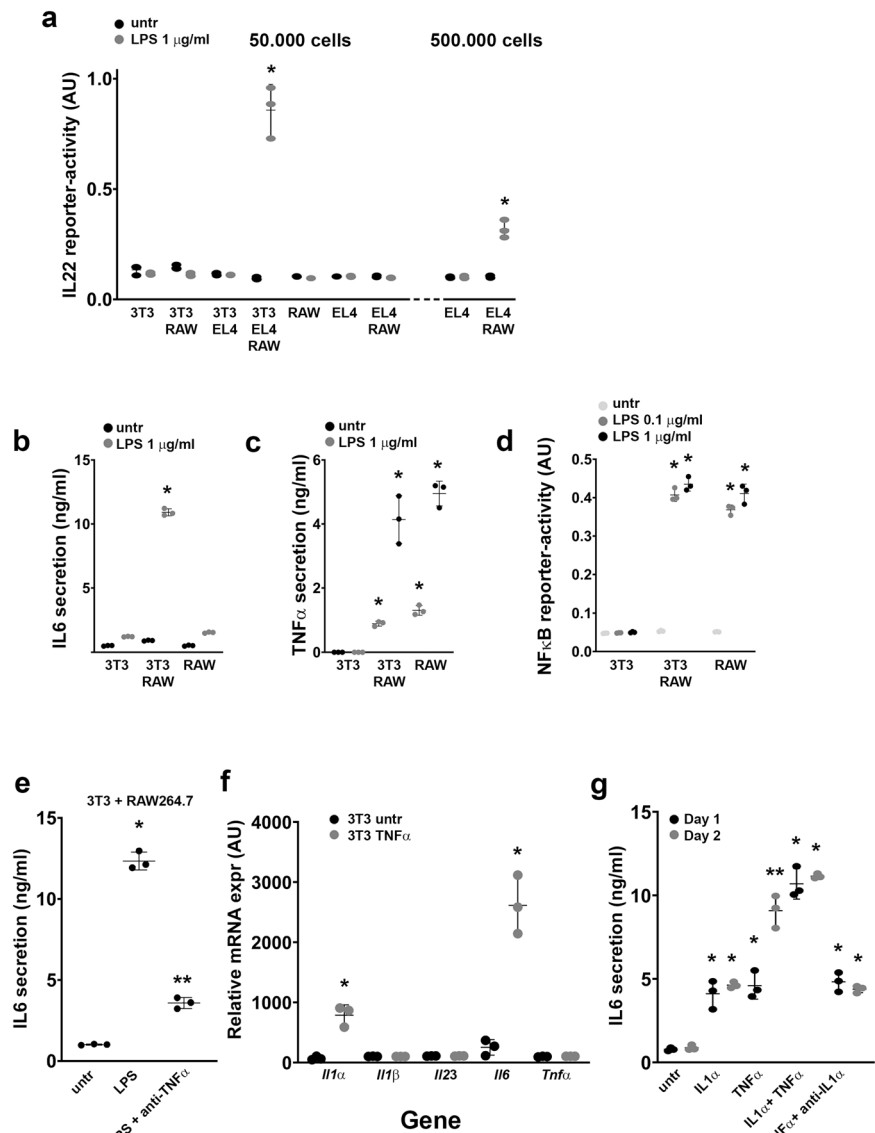

**Fig. 5 Fibroblasts enhance the macrophage-induced IL22 expression in EL4 T cells by enhanced IL6 and IL1α expression. a** IL22 reporter-activity of supernatants from indicated co-cultures of RAW264.7, NIH3T3, and EL4 cells stimulated with LPS (1 µg/ml) for 48 h. **b** IL6 ELISA on supernatants from indicated cultures of RAW264.7 and/or NIH3T3, stimulated with LPS (1 µg/ml) for 48 h. **c** TNFα ELISA on supernatants from indicated cultures of RAW264.7 and/or NIH3T3, stimulated with LPS (1 µg/ml) for 48 h. **d** NFκB reporter-activity of the RAW264.7 cells of indicated cultures of RAW264.7 and/or NIH3T3, stimulated with LPS (0.1 or 1 µg/ml) for 48 h. **e** IL6 ELISA on supernatants from co-cultures of RAW264.7 and NIH3T3, stimulated with LPS (1 µg/ml) for 48 h in the presence or absence of TNFα blocking antibody. **f** Cytokine mRNA expression in NIH3T3 after LPS (1 µg/ml) exposure for 24 h. **g** IL6 ELISA on supernatants from NIH3T3, stimulated with indicated cytokines (10 ng/ml) or blocking antibody for 24 or 48 h. All experiments were performed in triplicate with a minimum of three independent experiments. All data are shown as mean ± SD. For statistical analyses (**a**, **d** and **g**), Log2 transformed data were used in Welch and Brown–Forsythe tests followed by Dunnett's T3 multiple comparisons test. For comparison in **b**, **c**, **e** and **f**, we performed an unpaired t-test (two-tailed) with Welch's correction on Log2 transformed data for untreated versus LPS or TNFα treatment for each cell combination. *$P < 0.05$ was considered to indicate statistical significance. **$P < 0.05$ compared to LPS-stimulated samples (**e**) or to day 1 vs. day 2 (**g**).

we could not detect IL7 in macrophages, fibroblasts or T cells (Fig. S4), others have described that IL7 can be secreted by enterocytes and goblet cells[23–25]. To confirm these observations, we tested if IL7 is produced in ileum-derived organoids in response to any of the cytokines IL1β, IL17A, IL22, IFNω, IFNβ, IFNγ, or TNFα. Of these cytokines only IL22 (Fig. 7b) and IFNβ1, in a concentration-dependent manner (Fig. 7b, c), increased the expression of *Il7* mRNA. Moreover, quantification of IL7 immunostainings of 2D organoid cultures treated with IL22 and IFNβ1 (Fig. 7d) showed that IL7 protein expression increases (Fig. 7e) in concordance with the observed increase of IL7 mRNA after IL22 or IFNβ1 exposure.

IFNβ1 is known to be induced in intestinal cells by viruses[26], which suggests that the induction of IL7 by IFNβ1 is part of a viral route for IL22 induction. Expression analyses of the published data set GSE149312[27] shows that the infection of human intestinal organoids with SARS-CoV-2 can indeed induce the expression of *Il7* (Fig. 7f).

We found that IL7 can induce the expression of IL22 in EL4 T cells, however, IL6 co-exposure was still required for this (Fig. 4). Therefore, we tested if IFNβ1 can also induce the expression of IL6 in either organoids, RAW264.7 macrophages, NIH3T3 fibroblasts or EL4 T cells. We observed an induction of *Il6* in NIH3T3 fibroblasts after IFNβ1 exposure (Fig. 7g), but not

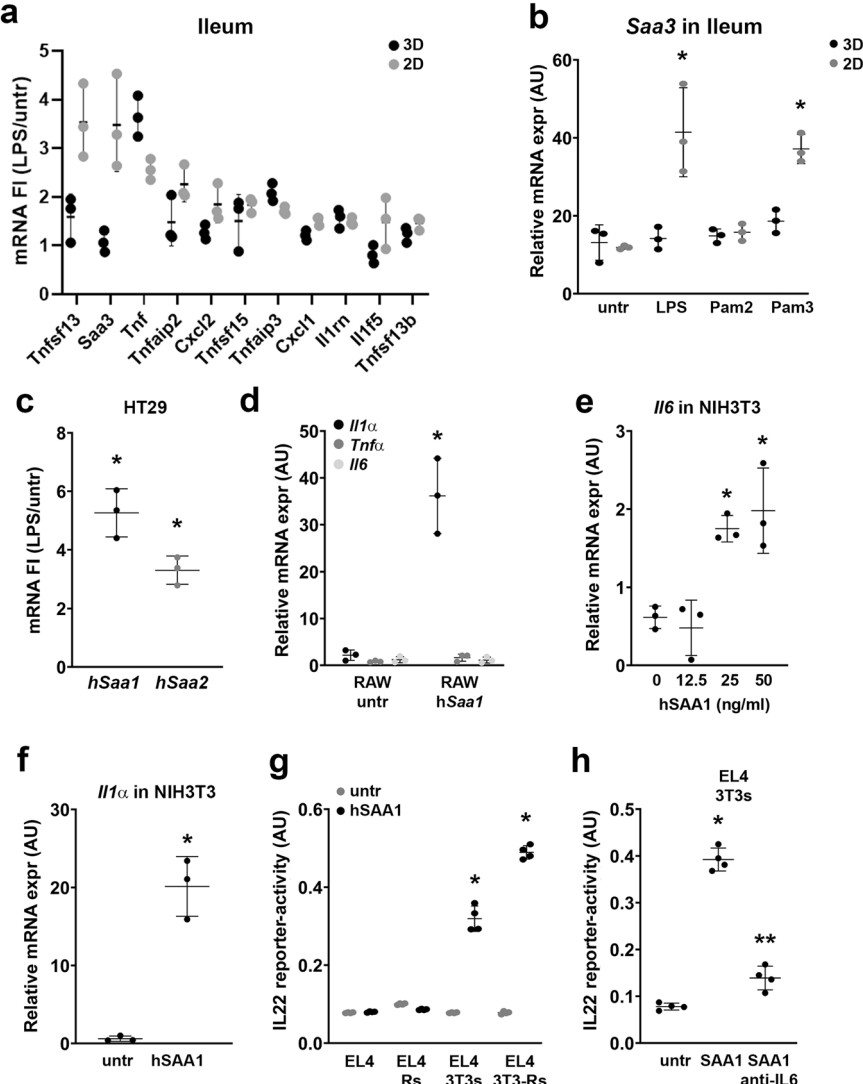

**Fig. 6 LPS-induced SAA3 from IEC induces IL22 in EL4 T cells through macrophage-derived IL1α and fibroblast-derived IL6 and IL1α. a** mRNA expression in LPS-stimulated 3D or 2D ileum-derived organoids of cytokine/chemokine selected genes (fold Increase). **b** *Saa3* mRNA expression in 3D or 2D ileum-derived organoids, stimulated with either LPS, Pam2, or Pam3 (1 μg/ml). **c** *hSaa1* and *hSaa2* mRNA expression in HT29 cells stimulated with LPS (1 μg/ml) for 24 h. **d** *Il1α*, *Tnfα*, and *Il6* mRNA expression in RAW264.7 cells after 24 h exposure to hSAA1 (10 ng/ml). **e** *Il6* mRNA expression in NIH3T3 fibroblasts after 24 h exposure to hSAA1 (10 ng/ml). **f** *Il1α* mRNA expression in NIH3T3 fibroblasts after 24 h exposure to hSAA1 (10 ng/ml). **g** IL22 reporter-activity of supernatants from EL4 cells exposed to supernatants of indicated cultures of RAW264.7 (Rs) and NIH3T3 (3T3s) cells stimulated with hSAA1 (10 ng/ml) for 48 h. **h** IL22 reporter-activity of supernatants from EL4 cells exposed to supernatants of NIH3T3 cells stimulated with hSAA1 (10 ng/ml) in the presence or absence of anti-IL6 blocking antibody. All experiments were performed in triplicate with a minimum of three independent experiments. All data are shown as mean ± SD. For statistical analyses (**a**, **b**, **e**, **g**, and **h**), Log2 transformed data were used in Welch and Brown–Forsythe tests followed by Dunnett's T3 multiple comparisons test. For comparison in **c**, **d** and **f**, we performed an unpaired *t*-test (two-tailed) with Welch's correction on Log2 transformed data.*$P < 0.05$ was considered to indicate statistical significance. **$P < 0.05$ compared to SAA1-stimulated samples (**h**).

in the other cell types (Fig. S5). Together, our data indicate that intestinal viral infections induce IL22 expression in T cells via IFNβ1-mediated IL7 production by epithelial cells and IL6 production in fibroblasts.

Because of the observed involvement of IL22 in antiviral pathways, we analyzed the expression of genes which are involved in viral infections and clearance (Table S1) and found that IL22 inhibits the expression of several viral entry receptors, such as *Ace2* (SARS-Cov-1 and 2), *Tmprss2* (SARS-Cov-1 and 2), *Dpp4* (MERS), *Tnfrsf14* (HSV), and *Cd46* (MV), while increasing the expression of antiviral proteins such as *Rsad2*, *Apobec1* and *3*, *Irf9*, *Mx1*, *Isg20*, and *AoS* (Fig. 7h). These findings suggest that IL22 modulates genes involved in viral entry and replication.

To test whether IL22 can indeed modulate viral infections, we used a GFP-expressing recombinant mouse hepatitis coronavirus (MHV-GFP)[28,29]. 2D mouse Ileum organoids were either untreated or treated for 1 day with IL22 (5 ng/ml) prior to viral infections with MHV-GFP. After 24 and 48 h, cells were fixed and stained for F-actin and nuclei (Fig. 7i) and the number of GFP-positive cells was determined (Fig. 7j). We observed a reduced number of MHV-positive cells 2 days of post MHV-GFP infection when organoids were exposed to IL22, indicating that IL22 can indeed reduce viral replication. Moreover, in IL22-treated organoids we observed mostly single MHV-GFP expressing cells, whereas in the untreated organoids most MHV-GFP-positive cells were clustered (Fig. 7i), suggesting that IL22 suppresses the infection of neighboring cells.

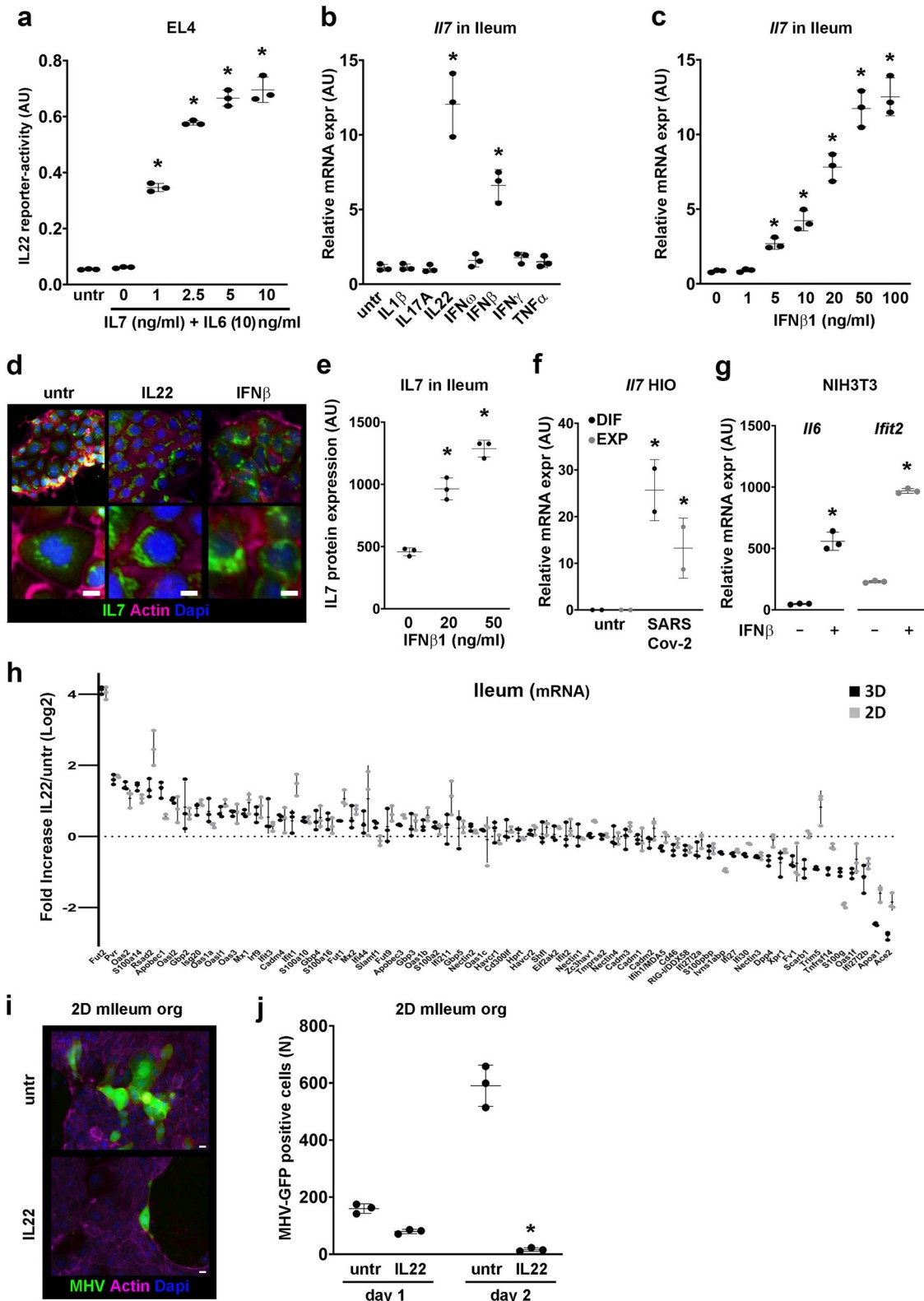

## Discussion

In this paper, we show for the first time that different immune cascades, providing crosstalk between enterocytes, macrophages, dendritic cells, fibroblasts, and T cells, are involved in the induction of IL22 in response to bacterial or viral infections (Fig. 8). Firstly, we show that the bacterial products LPS and Pam3 can induce the expression of SAA1 in enterocytes which can then induce the expression of IL1α in macrophages and fibroblasts. Furthermore, both SAA1 and IL1α can induce IL6 secretion from fibroblasts, which is required for IL22 secretion by T cells (Fig. 8). Secondly, LPS, Pam2, or Pam3 can directly induce the expression of TNFα, IL1α, IL1β, and IL6 in sampling dendritic cells or after damage of the intestinal epithelial layer[19], resulting in IL22 secretion in T cells (Fig. 8). In this

**Fig. 7 Virus-mediated expression of IFNβ1 induces IL22 via IL7 and IL6 produced by IEC and fibroblasts, respectively. a** IL22 reporter-activity of supernatants from EL4 cells stimulated with IL7 (indicated concentrations) and IL6 (10 ng/ml) for 48 h. **b** *Il7* mRNA expression in 3D ileum-derived organoids, stimulated with indicated cytokines (10 ng/ml) for 24 h. **c** *Il7* mRNA expression in 3D mouse organoids after IFNβ1 exposure for 24 h with indicated concentrations. **d** Immunostaining for IL7 (green), F-actin (magenta), or Nuclei (blue) in 2D ileum-derived organoids stimulated with IFNβ1 (20 ng/ml) or IL22 (2 ng/ml) for 24 h. Bars indicate 10 μm. **e** IL7 protein expression in 2D mouse organoids after IFNβ1 exposure for 24 h with indicated concentrations. **f** *Il7* mRNA expression in 3D human intestinal organoids, either differentiated (DIF) or undifferentiated (EXP), exposed to SARS-COV-2 for 60 h (GSE149312). **g** *Il6* and *Ifit2* mRNA expression in NIH3T3 fibroblasts stimulated with 20 ng/ml IFNβ1 for 24 h. **h** Microarray analysis of 2D and 3D mouse Ileum-derived organoids stimulated with IL22 (5 ng/ml) for 24 h. Gene expressions are indicated as Fold Increase (IL22/untr) in Log2 values. **i** MHV-GFP (green) expression, 2 days after infection, in 2D mouse ileum-derived organoids untreated or treated with IL22 (5 ng/ml), F-actin (magenta), or nuclei (blue). Bars indicate 10 μm. **j** Quantification of MHV-GFP positive cells in untreated and IL22 (5 ng/ml) treated 2D mouse ileum-derived organoids after 1 and 2 days of infection. All experiments were performed in triplicate with a minimum of three independent experiments. All data are shown as mean ± SD. For statistical analyses (**a**–**c** and **e**), Log2 transformed data were used in Welch and Brown–Forsythe tests followed by Dunnett's T3 multiple comparisons test. For comparison in **f**, **g**, and **j**, we performed an unpaired *t*-test (two-tailed) with Welch's correction on Log2 transformed data. *$P < 0.05$ was considered to indicate statistical significance.

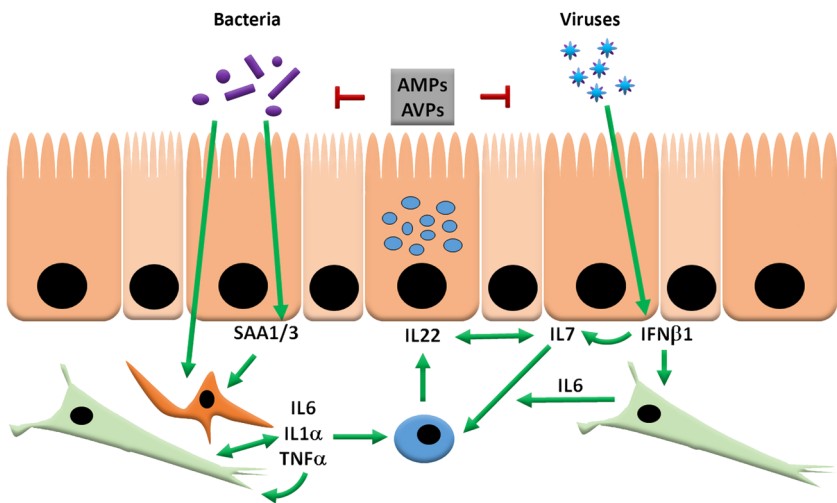

**Fig. 8 Model for bacterial and viral induced expression of IL22.** Bacterial-derived products, such as LPS and Pam3, can induce SAA1 (humans) or SAA3 (mouse), which can trigger dendritic cells or macrophages to produce IL1α and in turn, IL1α can induce the expression of IL6 in fibroblasts. In addition, SAA1 can also induce the expression of IL6 in fibroblasts in the absence of macrophages of dendritic cells. The secreted IL6 can then induce the expression of IL22 in T cells and this expression can be enhanced by IL7, which can be secreted by IEC. Moreover, the secretion of IL7 by IEC, can be enhanced by IL22, resulting in a positive feedback loop for IL22 induction and eventually giving higher expression of Reg3β and Reg3γ. Alternatively, LPS can directly activate sampling DCs and macrophages or activate these cells after intestinal damage, resulting in a strong immune response with high induction of IL6, IL1α, IL1β, TNFα, and subsequent IL22 induction, in which case TNFα enhances the IL6-induced IL22 production by T cells. A viral pathway involving IFNβ1 can also induce IL22 via IL7 produced by IEC and fibroblast- secreted IL6, after which anti-viral proteins (AVPs) can be induced by IL22.

pathway, IL22 secretion is greatly enhanced by positive feedback in two ways: TNFα increases the expression of IL6 in fibroblasts plus it enhances IL22 secretion by T cells in the presence of IL6. Thirdly, we provide evidence that virus-induced expression of IFNβ1 in intestinal cells leads to the expression of IL7 in enterocytes and IL6 in fibroblasts, resulting in IL22 secretion in T cells, and subsequent activation of antiviral pathways in IECs.

The two bacterial pathways leading to IL22 expression might reflect a basal versus an acute infection with SAA1 as a starting point for basal bacterial defense. By comparing germ-free mice and normal mice it has been elegantly shown that SAA3 (the mouse orthologue of human SAA1) is induced in the presence of gut microbiota[30], suggesting that SAA3 can indeed drive basal IL22 expression through a pathway involving dendritic cells, macrophages, fibroblasts, and T cells. Recently, it has also been shown that SAA3 can induce IL22 in neutrophils[7] or enhance IL6-induced IL22 secretion in Th17 cells[31], suggesting multiple routes through which mSAA3 or hSAA1 can enhance the expression of intestinal IL22 and AMPs.

IL22 has been shown to induce Reg3β and Reg3γ in intestinal cells in multiple studies[32–35], and here we found that IFNγ is also

able to induce the expression of these AMPs. Remarkably, we also found that IFNγ can inhibit the IL22-mediated upregulation of *Reg3β* and *Reg3γ*. These findings emphasize the importance of the specific cellular source of IL22, which in some conditions will be neutrophils, being strict IL22 producers[7], and in other conditions intraepithelial T cells, producing IL22 and IFNγ simultaneously (Fig. 3).

To our knowledge, the effect of IL7 on IL6-induced IL22 secretion has not been described before, although most studies on Th17, Th22, or ILC3 cells are performed in the presence of IL7, because of the growth-enhancing effect of IL7 on these T cells[5–8]. By using EL4 T cells, which do not require IL7 for growth, we could now discriminate between the IL7-induced T cell growth and IL7-enhanced IL22 secretion. We also show that IL7 expression in enterocytes can be enhanced by IL22, suggesting a positive feedback loop between IL22 and IL7. A recent study showed that a subset of IBD patients, which were not responsive to anti-TNFα treatment, had a higher IL7 pathway activity reflected by higher IL7 or IL7R expression[36]. This suggests that IL7 can indeed replace the function of TNFα with respect to IL22 production in the intestine. This, combined with

our observation that viruses can induce IL7 through IFNβ1, might also indicate that viral infections can affect the severity of IBD. Indeed metagenomic analyses of subjects with IBD reveal an interplay between intestinal viruses and bacteria[14,15].

Although IL22 is well-known for its role in bacterial defense, there is limited and conflicting data on the importance and regulation of IL22 in intestinal viral defense. For instance, influenza infection in mice is not enhanced in the presence of IL22-blocking antibodies[37], whereas IL22BP −/− mice show enhanced clearance of H1N1 virus[38]. In addition, MCMV replication in mice was enhanced when mice were treated with blocking antibodies for IL22[39]. IL22 can also protect against rotavirus infections as illustrated in vivo and in vitro by using either IL22 −/− mice, IL22 neutralizing antibodies and recombinant IL22[40–42]. Moreover, IL22 has also been implicated in direct inhibition of respiratory syncytial virus (RSV) autophagy[43] and dengue virus replication[44]. Consequently, a consensus is beginning to emerge that IL22 may exert antiviral control during infection. Indeed, our microarray analyses show increased expression of antiviral genes, such as Rsad2, Ifit1, 2, and 3 after IL22 exposure (Fig. 7h). We also observed reduced expression of the receptors Ace2, Tmprss2, Dpp4, Tnfrsf14, Aqp1, and Cd46 (Fig. 7h), which are involved in the entry of SARS-CoV and SARS-CoV-2[45,46], MERS[47], HSV[48], Dengue[49] and MV[50] respectively, indicating that IL22 can potentially inhibit viral entry. Indeed, infections of 2D mouse Ileum organoids with the mouse corona virus (MHV) could be reduced with an IL22 pretreatment (Fig. 7i, j). Especially the spreading of infections to neighboring cells appears to be blocked by IL22 (Fig. 7i).

Treatment of organoids with IL22 increases the expression of the fucosyltransferase 2 gene Fut2 (Fig. 7h) and in vivo treatment of mice with IL22 also results in higher fucosylation of enterocytes, most likely in a FUT2-dependent manner[51,52]. FUT2 can change the sugar moieties of extracellular surface proteins such as mucins[53,54] or histo-blood group antigen (HBGA)[55], which can either increase or decrease the entry of a specific virus[55], thereby changing the susceptibility to viral infections. Overall, viruses can induce the expression of IL22 through an IFNβ1–IL7–IL6-axis and successively, IL22 can help defend against viral infection on different levels by regulating 1) viral degradation, 2) viral entry, and 3) viral sensitivity through sugar modifications of extracellular proteins.

Collectively, here we provide fundamental knowledge that could help in designing selective strategies to improve innate intestinal defense processes preventing bacterial and viral infections and subsequent chronic inflammatory diseases, such as IBD.

## Methods

**Cell culture.** For all cultured cells, we used DMEM F12 + Glutamax medium (Gibco; 31331-028) supplemented with 10% Fetal Calf Serum (Gibco; 10082-147) or 5% for EL4 and RAW264.7 cells, 1% Pen/Strep (Gibco; 15070-063) and 1 mM sodium pyruvate (Gibco; 11360-039). JAWSII (ATCC® CRL-11904™) cells were grown in the presence of GMCSF (Peprotech; 315-03) 5 ng/ml and split 1:5 every 7 days. EL4 (ATCC® TIB-39™), RAW264.7 (ATCC® TIB-71™), JAWSII (ATCC® CRL-11904) and LR7[56] cells were split 1:30 every 3–4 days. NIH3T3 (ATCC® CRL-1658™) were split 1:10 every 3–4 days by scraping with a cell-scraper (Merck CLS3010-100EA).

**MHV-Srec GFP production.** Construction of MHV-2aGFPSRec, which contains an eGFP expression cassette between the 2a and the S genes at the position of the HE pseudogene, was described previously[28,29]. Briefly, LR7 cells were cultured in T75 bottles until a confluency of 80%. Subsequently, 100 µl of MHV-GFP recombinant virus suspension was added and incubated for 24 h. The next day, medium was collected and centrifuged for 5 min at 1200 rpm to remove LR7 cells. The virus containing medium was aliquoted and stored at −80 °C.

**Organoid isolation and culturing**

*Crypt isolation.* A female mouse C57/BL6, 6 months of age was sacrificed using carbondioxide. The intestine was removed and the ileum isolated. Villi were removed from the ileum by scraping with a scalpel and the tissue was cut into small pieces, transferred to a tube with ice-cold PBS and washed three times with PBS (spun down at 335 × g for 5 min between every wash step). A Polter–Elvehjem tube was then used to fragment the tissue and the cell suspension was transferred to a tube with ice-cold PBS, washed three times with PBS and subsequently filtered through a 100 µm filter. The now isolated crypts were spun down at 335 × g for 5 min at room temperature and supernatant was removed. The crypts were resuspended in Matrigel (Corning; 356231), and drops of 25 µl suspension were added to each well (Corning Costar). The plates were then incubated for 15 min at 37 °C until the droplets had solidified. One milliliter of organoid culture medium supplemented with 10% FCS, 10 µM Y27632 (Selleckchem; S1049), CHIR 4,3 µM, (Sigma; SML 1046) and 2-Propylpentanoic acid (Sigma; P6273; 1:8000). After 3 days of incubation at 37 °C and 5% CO₂, medium was replaced by normal culture medium: DMEM/F12 medium containing GlutaMAX supplement, 1 mM sodium pyruvate, MEM Non-Essential Amino Acids, 100U/ml Penicillin-Streptomycin, R-spondin 1 (homebrew), WNT (homebrew), and the BMP4 inhibitor DMH1 0.5 µg/ml (Sigma; D8946) to stimulate organoid formation. These organoids were passaged once a week in a 1:4 ratio.

For the secretion assays, stainings and QPCR analysis, the organoids were plated according to our 2-dimensional method. Twenty-four wells plates were coated with a thin layer of 3 µg/ml rat tail Collagen I (IBIDI GmbH; 50201). After solidification of the Collagen I coating, a suspension of organoids in medium supplemented with 10% FCS, 10 µM Y27632 (Selleckchem; S1049), CHIR 4,3 µM, (Sigma; SML 1046) and 2-Propylpentanoic acid (Sigma; P6273; 1:8000) was added to each well. The organoids were grown for 2 days at 37 °C and 5% CO₂. The use of mouse intestine was approved by the local committee for care and use of laboratory animals at Utrecht University. Only surplus mice were used for this study.

**Murine bone marrow-derived dendritic cells and intraepithelial lymphocytes.** Bone marrow-derived dendritic cells (DC) were isolated from C57BL/6 mice (female 6 months) and cultured in presence of GM-CSF for 6 days according to the methods described in Lutz et al.[57]. Medium was refreshed after 3 days. DCs were collected and stimulated with 10 ng/ml LPS (Sigma) for 24 h in a 12-wells plate (Greiner-bio-one, 1 × 10⁶ cells/well). Supernatant was collected and stored at −80 °C until use.

Small intestine of female C57BL/6 mice (7–10 weeks old) was removed and flushed with phosphate buffered saline (PBS, Biowhittaker) to remove luminal contents. After removing fat tissue and Peyer's patches, the intestines were cut open longitudinally and the epithelial layer was scraped off. Tissue was collected in RPMI (supplemented with 10% FCS and 1% penicillin/streptomycin) containing 1 mM dithiothreitol (DTT) and incubated for 40 min with gentle rotation. Tissue fragments were washed once, vortexed to obtain single cells and run through a nylon wool column to remove excess epithelial cells and non-T immune cells. The eluates were washed once and a density gradient centrifugation was performed using 40/80% percoll (GE Healthcare, Bio-sciences AB, Sweden). The cell layer at the 40/80% interface was harvested. These freshly isolated IEL were stimulated with BMDC LPS supernatant at a density of 1 × 10⁵ cells/well in a 96-wells round bottom plate (Greiner). Culture supernatant was harvested after 24 h and used to stimulate intestinal organoids. The use of mouse intestine was approved by the local committee for care and use of laboratory animals at Utrecht University. Only surplus mice were used for this study.

**Cell stimulation.** JAWSII and RAW264.7 cells (50,000 cells per well) were seeded in the presence of E. coli-derived B4 LPS (Sigma; L4391) 0.2 µg/ml for 18 h and supernatant was collected the next day. Subsequently, EL4 cells were seeded in the presence of LPS, JAWSII supernatant or cytokines (Peprotech) as indicated and incubated for 18 h before cytokine and QPCR analysis.

NIH3T3 fibroblast were seeded in 12-wells plates, 300,000 cells/well/3 ml and allowed to adhere for 24 h. Next, when indicated, we added 50,000 RAW264.7 cells and/or 50,000 EL4 cells. Subsequent stimulations were performed for indicated time points and concentrations.

Recombinant mSAA3 was produced in HEK293T cells and purified by Ni-beads. Stimulations with hSAA1 (Peprotech 300-53) were performed in the presence of Polymyxin B (Merck 1547007-200MG), which sequesters potential LPS contamination. Recombinant mIL22BP (RnDsystems 2376-BP-025).

Blocking antibodies that were used are: anti-mIL6 (Peprotech 500-P56), anti-mIL22 (Peprotech 500-P223), anti-mTNFβ (Peprotech 500-P64), anti-mIL9 (Peprotech 500-P59), anti-mIL7 (Peprotech 500-P57), anti-mIFNγ (Peprotech 500-P119), anti-mIFNγR1 (RnDsystems MAB10262), anti-IFNγR2 (RnDsystems MAB773), and anti-mIL1α (Peprotech 500-P51).

**RNA isolation and real-time PCR quantification.** Briefly, RNA from each sample was isolated (Zymo Research; R1055) and 200 ng of RNA was used for cDNA synthesis by reverse transcription (Bio Rad; 170-8891). Real-time PCR reactions included 5 µl of diluted RT product (1:6 dilution), 10 µl FAST SYBR Green buffer (Applied Biosysems; 4385614), 4.8 µl H₂O and 0.5 µM forward and 0.5 µM reverse primer. Reactions were incubated in an Applied Biosystems 7500 Fast Real-Time PCR system in 96-well plates. The primers used are described in Table S2. All

QPCR gene expression data was first normalized to reference gene *Hprt*, subsequently plotted relative to control.

**Microarray**. One hundred nanogram of RNA was used for Whole Transcript cDNA synthesis (Affymetrix, inc., Santa Clara, USA). Hybridization, washing and scanning of Affymetrix GeneChip Mouse Gene 1.1 ST arrays was carried out according to standard Affymetrix protocols. All arrays of the small intestine were hybridized in one experiment. Arrays were normalized using the Robust Multi-array Average method[58,59]. Probe sets were assigned to unique gene identifiers, in this case Entrez IDs. The probes on the Mouse Gene 1.1 ST arrays represent 21,213 Entrez IDs. Array data were analyzed using an in-house, on-line system[60].

**IL22, IL6, TNFα, and INFy ELISA**. Supernatants of exposed cells were collected and diluted 1:1 in ELISA dilution buffer according to manufacturer's protocol (R&D; DY582, DY206-05, and DY485). Briefly, plates were coated with capture antibody for 18 h, blocked for 1 h, washed 3× and exposed for 2 h to 100 μl diluted supernatant from exposed cells. Then plates were washed 3× and detection antibody was added for another 2 h after which the plates were washed again for 3× and streptavidin-HRP was added for 30 min. Plates were washed again and HRP substrate (Sigma; T4444) was added for 10–30 min after which the reaction was stopped by adding 50 μl of 1 M $H_2SO_4$. Colorimetric determination was performed in an ELISA plate reader at 450 nm wavelength.

**Immunofluorescent staining**. Cultured organoids were fixed with 4% formaldehyde for 15 min and permeabilized with 0.5% Triton-X100 for 5 min and subsequently washed twice with PBS. These organoids were incubated with either anti-IL7 antibody (Peprotech; 500-P57) for 2 h at 37 °C. Alexa Fluor 488 goat anti-rabbit (Life Technologies) and Alexa Fluor 488 donkey anti-mouse (Life Technologies) were used as secondary antibodies. Actin and nuclei were stained using Rhodamine Phalloidin (Life Technologies) and Hoechst 33342 (Sigma-Aldrich; 14533), respectively. Images were acquired on a Olympus IX71 fluorescence microscope using CellSens software (Olympus Corporation).

**Statistics and reproducibility**. All data is shown as mean ± SD. For multiple conditions, measurements were analyzed on Log2 transformed data by a Welch and Brown–Forsythe test followed by Dunnett's T3 multiple comparisons test, $P < 0.05$ was considered to indicate statistical significance. For comparison of two groups, we performed an unpaired *t*-test (two-tailed) with Welch's correction on Log2 transformed data. All experiments were performed in triplicate and all experiments were executed at least three times, except for the array data, which were performed only once in triplicate.

**Reporting summary**. Further information on research design is available in the Nature Research Reporting Summary linked to this article.

## Data availability
The authors declare that the data supporting the findings of this study are available within the article, the Supplementary Data 1–7, Supplementary Figs. S1–4, and Supplementary Tables 1 and 2, or are available on reasonable request. The microarray data discussed in this publication have been deposited in the Gene Expression Omnibus database GSE171798.

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

## Acknowledgements

The work has been partly funded by Dutch funding NWO-SIA via RAAK PRO project 2014-1-88PRO entitled 2RealGuts.

## Author contributions

J.P.T.K. designed, performed, and analyzed the data and wrote the paper. M.B.S. performed IEL and DC isolations from mice. K.S. provided technical support and S.V. and R.P. coordinated the project. F.K. coordinated the work on viral infections that were performed by A.vV. and C.dH. provided us with the MHV-GFP. All authors reviewed and approved the final version of the manuscript.

## Competing interests

The authors declare no competing interests.
