## [Peer Review File · Communications Biology]

Reviewers' comments:

Reviewer #1 (Remarks to the Author):

In this study, ten Klooster et al. report that bacterial and viral stimuli can drive IL22 production by T cells via distinct signaling pathways, and that IL22 can stimulate expression of anti-microbial and antiviral factors. Overall, this study does a nice job of methodically going through IL22 induction in T cells by bacterially-stimulated dendritic cells and macrophages, using immortalized cell lines and primary cells. Data regarding IL-6/TNF α mediated induction of IL22 production is also nicely done. Figure 7 is somewhat weaker and less convincing, as the dots are much less clearly connected following viral infection/stimuli, but this data is still of interest. Key issues in this paper are a current lack of transparency regarding raw data and N, inadequate use and description of appropriate statistical analysis, and missing details on methods.

Major comments:

1. As shown in Figure 1 and in the legend, it is not entirely clear whether these are all qPCRs or organoids or some are the EL4 or JAWSII, etc, cells themselves. Are the "JAWSII" and "RAW264.7" groups just these unstimulated cells incubated with organoids? Showing -/+ for each individual factor beneath each bar would make this figure much easier to interpret.

2. Both number of replicates and number of experiments are not shown in the legends, and the use of bar graphs instead of individual datapoints also makes this information hard to decipher. Please convert column bar graphs into scatter plots throughout for added transparency, and include N as well as number of independent experiments throughout.

3. Use of t-tests is likely inappropriate for these datasets, as they are unlikely to be normally distributed. Kruskal-Wallis followed by Dunn's multiple comparisons test or something similar, reflecting the likely nonparametric nature of the data, is needed. Please also add the statistical analysis approach used to each legend for clarity.

4. Why was IFN-lambda not included in the analyses in Figure 2B and 7B, as this is a known important stimulatory cytokine in the intestinal epithelium in response to viral infections? SARS-CoV-2 also induces IFN-lambda, so this could be a key regulator for IL-7 expression.

5. Can the authors comment on the use of physiologically-relevant levels of cytokines and/or SAA1 in their in vitro studies? Also, what was the source of these, especially SAA1 – was bacterial contamination a consideration?

6. There is lots of data about IL22 and regulation of enteric rotavirus infection (PMID 26006013, 28322925, 25395539). This should be discussed.

Minor comments:

1. Careful editing is needed throughout – for example, in the introduction IBD, Inflammatory Bowel Disease, is designated "Intestinal Bowl Disease". Transcript versus protein annotations of genetic factors are not accurate in text or figures.

2. It is very difficult to read anything in Figure 2A. Please reformat so gene names are readable.

3. The following statement, "This suggests that the increase of bacteria results in an increase of SAA3 expression in vivo and that a basal level of SAA3 is always present in the small intestine." seems over-reaching, as there are other differences between duodenum and ileum independent, or related to, bacterial load that could be at play. Should support with orthologous data or literature.

4. The statement, "These findings suggest that IL22 down-regulates both genes involved in viral entry and replication." Seems to be an overstatement, as there are entry factors and ISGs up- and down-regulated by IL22 in this system. Perhaps "modulated" would be more appropriate?

Reviewer #2 (Remarks to the Author):

Reviewer Comments

The manuscript submitted by ten Klooster et al shed light how enterocyte, Fibroblast, and myeloid cell function synergizes to IL22 induction and effective clearance of bacterial and viral infection. The author developed very nice invitro intestinal organoid approach to determine cell specific IL22 expression and mediated Reg3 β , Reg3 γ antimicrobial peptide induction. The author data described that more than 50 targeted gene upregulated with LPS stimulation in figure 4A. Later, the author attempts to induced LPS to myeloid cells and use supernatant to stimulate EL4 T Cells. With lack of enough genetic validation of the data findings of the manuscript weakens. Also, in recent years, LPS induced IL23, IL1 β , IL6 and TNF α secretion from dendritic cells and macrophages shown to induce IL22 from ILC3 cells is very well established. Infect, IL23a KO mice has been shown similar and mirror phenotype as IL22KO mice (Zheng Y, Nature. 2007; 445:648-651; Buonocore S, Nature. 2010; 464:1371-1375). Although the author entirely excludes the role of IL23/ILC3 in their manuscript. Please find the specific comments for the manuscript.

Major

1. In Fig 2G the author has shown presence of anti-IL22 antibody results to lower Reg3g expression. Anti-IL22 neutralizing antibody information (clone) and their condition is not describing in the manuscript. IL22 function led to development and maintenance of MALT and anti-IL-22 neutralizing antibody could lose colon-patches. Therefore, blocking of IL22 receptor, IL22Ra shown will be much effective approach by blocking anti-IL22Ra blocking antibody. I will recommend to the authors using ileum organoids from IL22Ra KO or blocking with Anti-IL22Ra blocking antibody will strength and further validate their findings.
2. Figure 3A, 3B the author stimulated IEL organoid culture with DC-LPS supernatant. Here the author should perform immune profiling that what kind of cells including, ILC3 cell, T Cells, NKT cell or $\gamma\delta$ T cell in IEL is responsible for secreting IL22 cytokine by FACS staining. The author should use IEL from Rag KO mice to verify that if in their system IL22 induction is ILC3 independent.
3. Figure 4B, the author should explain the rationale behind using HEK Blue-IL22 reporter cells to detect IL22 reporter activity. For example, IL23 receptor is do not expresses on HEK cell and therefore, the assay concluded that IL23 has no role in IL22 induction. Figure 5D, no detailed information provide about which cell has been used for NF-kB reporter activity.
4. The author did not make any attempt to infect their organoid culture system with viral infection to validate whether IL22 has anti-viral effect, and IFN β 1 and IL7 synergies the effect, figure7. This will certainly strength their manuscript. Although, the author used previously published data to establish their hypothesis that IL22 is important in viral clearance and examine viral related genes which is not enough, in figure 7H.

Minor

5. Figure 5 A the author has cultured at least three cell line together and they did not mention the culture conditions specific ratio etc.
6. RT PCR stimulation given for 18 hours, as described in methods and in entire manuscript although 1-3 hour stimulation is enough to detect transcript signal.

Reviewers' comments:

Reviewer #1 (Remarks to the Author):

In this study, ten Klooster et al. report that bacterial and viral stimuli can drive IL22 production by T cells via distinct signaling pathways, and that IL22 can stimulate expression of anti-microbial and antiviral factors. Overall, this study does a nice job of methodically going through IL22 induction in T cells by bacterially-stimulated dendritic cells and macrophages, using immortalized cell lines and primary cells. Data regarding IL-6/TNF α mediated induction of IL22 production is also nicely done. Figure 7 is somewhat weaker and less convincing, as the dots are much less clearly connected following viral infection/stimuli, but this data is still of interest. Key issues in this paper are a current lack of transparency regarding raw data and N, inadequate use and description of appropriate statistical analysis, and missing details on methods.

Dear Reviewer, thank you for carefully reading our manuscript and for your feedback

Major comments:

1. As shown in Figure 1 and in the legend, it is not entirely clear whether these are all qPCRs or organoids or some are the EL4 or JAWSII, etc, cells themselves. Are the "JAWSII" and "RAW264.7" groups just these unstimulated cells incubated with organoids? Showing +/- for each individual factor beneath each bar would make this figure much easier to interpret.

In Figure 1 all QPCR data is performed on organoids exposed to supernatants of the different cell types and treatments. However, for clarity we have adjusted the figure as you have suggested.

2. Both number of replicates and number of experiments are not shown in the legends, and the use of bar graphs instead of individual datapoints also makes this information hard to decipher. Please convert column bar graphs into scatter plots throughout for added transparency, and include N as well as number of independent experiments throughout.

Although we had stated in the Methods section, subsection "Statistics", that all experiments were performed 3 times in triplicate, we have now included this also in the figure legends.

"All data were performed in triplicate at with a minimal of 3 independent experiments."

3. Use of t-tests is likely inappropriate for these datasets, as they are unlikely to be normally distributed. Kruskal-Wallis followed by Dunn's multiple comparisons test or something similar, reflecting the likely nonparametric nature of the data, is needed. Please also add the statistical analysis approach used to each legend for clarity.

Indeed, most data are not normally distributed, such as data from QPCR and ELISA. To adjust this, we first performed a Log2 transformation and subsequently performed either a Welch and Brown-Forsythe test followed by Dunnett's T3 multiple comparisons test or an

unpaired t-test (two-tailed) with Welch's correction. This is noted in the Material and Methods section and also included in the figure legends.

“Statistical analyses

All data is shown as mean SD. For multiple conditions, measurements were analyzed on Log2 transformed data by a Welch and Brown-Forsythe test followed by Dunnett's T3 multiple comparisons test, P<0.05 was considered to indicate statistical significance. For comparison of 2 groups, we performed an unpaired t test (two-tailed) with Welch's correction on Log2 transformed data. All experiments were performed in triplicate and all experiments were executed at least three times, except for the array data, which were performed only once in triplicate.”

4. Why was IFN-lambda not included in the analyses in Figure 2B and 7B, as this is a known important stimulatory cytokine in the intestinal epithelium in response to viral infections? SARS-CoV-2 also induces IFN-lambda, so this could be a key regulator for IL-7 expression.

IFN-Lambda was initially excluded because our array data show that this protein is already expressed by untreated organoids. However, we appreciate the suggestion and tested the effect of IFN-L2 and L3 on organoids and observed no increase on Reg3 β , Reg3 γ and IL7 mRNA expression (data not shown).

5. Can the authors comment on the use of physiologically-relevant levels of cytokines and/or SAA1 in their in vitro studies? Also, what was the source of these, especially SAA1 – was bacterial contamination a consideration?

According to literature, in normal conditions, SAA concentration in serum is approximately 100-1000 ng/ml (Curr Med Chem. 2016 May;Structure and Expression of Different Serum Amyloid A (SAA) Variants and their Concentration-Dependent Functions During Host Insults. Mieke De Buck and Jo Van Dammea). However, during an acute-phase reaction, the concentration can rise to 1 mg/mL or even higher. So even basal levels in serum are in the range of the concentrations used in our experiments.

Regarding the source of mSAA3 and hSAA1, they were produced in HEK cells and E coli, respectively. The experiments with hSAA1 were therefore performed in the presence of Polymyxin B sulfate to remove potential LPS contaminations. The following text was included in the paper:

“NIH/3T3 Fibroblasts were seeded in 12 wells plates, 300.000 cells/well/3 ml and allowed to adhere for 24h. Subsequent stimulations were performed for indicated times and concentrations. Recombinant mSAA3 was produced in HEK293T cells and purified using Ni-beads. Stimulations with hSAA1 (Peprotech 300-53) were performed in the presence of Polymyxin B (Merck 1547007-200MG), which sequesters potential LPS contamination.”

6. There is lots of data about IL22 and regulation of enteric rotavirus infection (PMID 26006013, 28322925, 25395539). This should be discussed.

These are indeed very nice papers showing the involvement of IL22 in defense against rotavirus infections. We have added the following sentence in the discussion.

“IL22 can also protect against rotavirus infections which is nicely illustrated in vivo and in vitro by using either IL22 ko mice, IL22 neutralizing antibodies and recombinant IL22.”

Minor comments:

1. Careful editing is needed throughout – for example, in the introduction IBD, Inflammatory Bowel Disease, is designated “Intestinal Bowl Disease”. Transcript versus protein annotations of genetic factors are not accurate in text or figures.

We have adjusted the suggested text throughout the manuscript.

2. It is very difficult to read anything in Figure 2A. Please reformat so gene names are readable.

We have included a larger version of Figure 2A as supplemental Figure S3.

3. The following statement, “This suggests that the increase of bacteria results in an increase of SAA3 expression in vivo and that a basal level of SAA3 is always present in the small intestine.” seems over-reaching, as there are other differences between duodenum and ileum independent, or related to, bacterial load that could be at play. Should support with orthologous data or literature.

We agree with the reviewer stating that this conclusion might be over-reaching. In addition there is ample published data showing that the microbiome induces the expression of SAA in intestinal cells, which we already referred to in our discussion. Therefore we removed this figure from the manuscript.

4. The statement, “These findings suggest that IL22 down-regulates both genes involved in viral entry and replication.” Seems to be an overstatement, as there are entry factors and ISGs up- and down-regulated by IL22 in this system. Perhaps “modulated” would be more appropriate?

The statement has been changed into:

“These findings suggest that IL22 modulates genes involved in viral entry and replication.”

Dear Reviewer, thank you for carefully reading our manuscript and for your feedback.

Reviewer #2 (Remarks to the Author):

Reviewer Comments

The manuscript submitted by ten Klooster et al shed light how enterocyte, Fibroblast, and myeloid cell function synergizes to IL22 induction and effective clearance of bacterial and viral infection. The author developed very nice in vitro intestinal organoid approach to determine cell specific IL22 expression and mediated Reg3 β , Reg3 γ antimicrobial peptide induction. The author data described that more than 50 targeted gene upregulated with LPS stimulation in figure 4A. Later, the author attempts to induced LPS to myeloid cells and use supernatant to stimulate EL4 T Cells. With lack of enough genetic validation of the data findings of the manuscript weakens. Also, in recent years, LPS induced IL23, IL1 β , IL6 and TNF α secretion from dendritic cells and macrophages shown to induce IL22 from ILC3 cells is very well established. Infect, IL23a KO mice has been shown similar and mirror phenotype as IL22KO mice (Zheng Y, Nature. 2007; 445:648–651; Buonocore S, Nature. 2010; 464:1371–1375). Although the author entirely excludes the role of IL23/ILC3 in their manuscript. Please find the specific comments for the manuscript.

Major

1. In Fig 2G the author has shown presence of anti-IL22 antibody results to lower Reg3g expression. Anti-IL22 neutralizing antibody information (clone) and their condition is not describing in the manuscript. IL22 function led to development and maintenance of MALT and anti-IL-22 neutralizing antibody could lose colon-patches. Therefore, blocking of IL22 receptor, IL22Ra shown will be much effective approach by blocking anti-IL22Ra blocking antibody. I will recommend to the authors using ileum organoids from IL22Ra KO or blocking with Anti-IL22Ra blocking antibody will strength and further validate their findings.

The used antibodies are described in Methods, subsection “Cell stimulation”, last paragraph.

Regarding the use of neutralizing antibodies for IL22RA1 instead of anti-mIL22, we decided on blocking IL22, because blocking IL22RA1 would potentially also block IL20 and IL24 signaling (Dumoutier, L. et al. (2001) J. Immunol; Wang, M. et al. (2002) J. Biol. Chem.). However, performing the same experiment and replacing anti-mIL22 by IL22BP, a soluble IL22 receptor known for its inhibitory effect on IL22, gave the same results. Although we did not add a new figure to the manuscript, we have included it as data not shown.

“or presence of blocking antibodies for IL22 and IFN γ and their receptors (data not shown).”

2. Figure 3A, 3B the author stimulated IEL organoid culture with DC-LPS supernatant. Here the author should perform immune profiling that what kind of cells including, ILC3 cell, T

Cells, NKT cell or $\gamma\delta$ T cell in IEL is responsible for secreting IL22 cytokine by FACS staining. The author should use IEL from Rag KO mice to verify that if in their system IL22 induction is ILC3 independent.

We indeed tried to stain for mIL22 in the isolated IEL cells but were unsuccessful with any of the commercially available antibodies against mIL22 we tested. Most published IL22 FACS analyses data we found were performed on human cells. This is also one of the reasons why others have developed IL22-promoter-YFP mice to identify IL22-expressing cells in the intestine. Unfortunately, we do not have access to these mice. In addition, there is an increasing number of publications suggesting that ILCs have a high plasticity regarding their functions and cytokine productions (Nature Reviews Immunology 27 February 2020; Plasticity of innate lymphoid cell subsets. Suzanne M. Bal, Korneliusz Golebski & Hergen Spits). The purpose of our experiment was to show that freshly isolated intestinal residing T cells are also responsive to IL6 and TNF α , similar to EL4 cells which we used as a model system. For these reasons we did not claim, in our manuscript, any specific IEL to be the unique source of IL22 and most likely, different sub-populations are involved.

3. Figure 4B, the author should explain the rationale behind using HEK Blue-IL22 reporter cells to detect IL22 reporter activity. For example, IL23 receptor is do not expresses on HEK cell and therefore, the assay concluded that IL23 has no role in IL22 induction.

The HEK Blue-IL22 reporters are used to determine the levels of IL22 produced by EL4 cells, which do have the IL23R. Therefore, the presence of a IL23R on HEK cells is not relevant for the conclusion that IL23 is not involved in the production of IL22 by EL4 T cells.

Figure 5D, no detailed information provide about which cell has been used for NF- κ B reporter activity.

The NF κ B-reporter activity was present only in the RAW264.7 cells. Although stated in the main text, this was indeed not clear from the legends and has now been included in the legends for Figure 4.

"NF κ B reporter-activity of the RAW264.7 cells of indicated cultures of RAW264.7 and/or NIH3T3, stimulated with LPS (0.1 or 1 μ g/ml) for 48h."

4. The author did not make any attempt to infect their organoid culture system with viral infection to validate whether IL22 has anti-viral effect, and IFN β 1 and IL7 synergies the effect, figure7. This will certainly strength their manuscript. Although, the author used previously published data to establish their hypothesis that IL22 is important in viral clearance and examine viral related genes which is not enough, in figure 7H.

We agree that performing viral infections would be a great improvement of the manuscript. Therefore, we performed viral infections using the mouse corona virus (MHV-GFP) in collaboration with the group of Frank van Kuppeveld (presented in figure 7I and 7J) in which we observed that pretreatment of mouse Ileum organoids with IL22 reduces the

number of MHV-GFP infected intestinal cells.

Minor

5. Figure 5 A the author has cultured at least three cell line together and they did not mention the culture conditions specific ratio etc.

We have now included the culture conditions in Methods, subsection "Cell stimulation":

"NIH/3T3 Fibroblast were seeded in 12 wells plates, 300.000 cells/well/3 ml and allowed to adhere for 24h. Next, when indicated, we added 50.000 RAW264.7 cells and/or 50.000 EL4 cells. Subsequent stimulations were performed for indicated times and concentrations."

6. RT PCR stimulation given for 18 hours, as described in methods and in entire manuscript although 1-3 hour stimulation is enough to detect transcript signal.

We performed the QPCR analyses after 18 hours, because that is also the time point at which we collected the supernatants for subsequent stimulations and protein analyses.

REVIEWERS' COMMENTS:

Reviewer #2 (Remarks to the Author):

The authors have addressed my main concerns and the manuscript is substantially clearer now.

Reviewer #3 (Remarks to the Author):

The revised version of the manuscript submitted by ten Klooster looks significantly improved. The author has adequately taken care of my comments and explained my raised concerns to improve.

My major concern was that the author did not attempt to infect their organoid culture system with a viral infection to validate whether IL22 has an anti-viral effect. In the revised version of the manuscript, the author has added pretreatment of mouse Ileum organoids with IL22 and shown IL22 reduces the number of MHV-GFP infected intestinal cells.

I will recommend for acceptance.